# Sensitivity of tunable infrared laser spectroscopic measurements of $\Delta'^{17}O$ in $CO_2$ to analytical conditions

David Bajnai[1], Vincent J. Hare[2,3]

[1]Department of Geochemistry and Isotope Geology, Geoscience Center Göttingen, University of Göttingen, Göttingen, Germany

[2]Stable Light Isotope Laboratory, Department of Archaeology, University of Cape Town, Cape Town, South Africa

[3]Node for Isotope Biogeochemistry, BIOGRIP, University of Cape Town, Cape Town, South Africa

*Correspondence to*: David Bajnai (david.bajnai@uni-goettingen.de)

**Abstract.**

Triple oxygen isotope ($\Delta'^{17}O$) measurements of $CO_2$ are increasingly used in atmospheric and paleoenvironmental research, supported by the rise of tunable infrared laser direct absorption spectroscopy (TILDAS) as a cost- and time-effective method for quantifying rare isotopologues in $CO_2$. In this study we use data from two TILDAS instruments (University of Göttingen and University of Cape Town) to evaluate how the stability of analytical conditions, such as temperature, optical cell pressure, and the analyte's $CO_2$ concentration influences measurement repeatability. We identify the mismatch in $CO_2$ concentration between sample and working gas as the most significant factor affecting the repeatability of $\Delta'^{17}O$ measurements. The resulting scale-offset effect can amount to several ppm per $\mu mol\ mol^{-1}$ mismatch, depending on instrumental configuration. Applying empirical corrections for such offsets significantly improves repeatability. We show that maintaining electronics temperature, cell temperature, and cell pressure stability within $\pm 0.1$ K, $\pm 1$ mK, and $\pm 10$ Pa, respectively, across the cycles of a replicate measurement is sufficient to avoid any resolvable impact on the internal error. Over multi-week timescales, drift in measured isotope ratios is primarily driven by variations in the temperature of electronic components, while fluctuations in cell temperature, pressure, and analyte concentration exert smaller but detectable influences. We conclude with practical recommendations for achieving high precision triple oxygen isotope measurements with TILDAS, emphasizing that continuous monitoring and reporting of analytical conditions is essential.

## 1 Introduction

Oxygen has three stable isotopes, with 99.7% as $^{16}O$, 0.2% as $^{18}O$, and only about 0.04% as $^{17}O$. Measurements of the $^{18}O/^{16}O$ ratio in solids, liquids, and gases has proved invaluable across nearly every field of Earth science. Additional measurements of the $^{17}O/^{16}O$ ratios, known as triple oxygen isotope analyses, provides additional information on fractionation mechanisms and the mixing and transport processes that drive variations in oxygen isotope ratios (Pack and Herwartz, 2014; Young et al., 2002).

Mass-dependent fractionation processes result in a correlation between the $\delta^{17}O$ and $\delta^{18}O$ values, following slopes of approximately 0.5 (Craig, 1957; Matsuhisa et al., 1978; Young et al., 2002). The exact slope of the correlation depends on the specific mass dependent fractionation process and temperature. The $\Delta'^{17}O$ notation expresses deviations in the relative abundances of oxygen isotopes from a reference line:

$$\Delta'^{17}O = \ln(\delta^{17}O + 1) - \lambda_{RL} \cdot \ln(\delta^{18}O + 1) - \gamma_{RL}$$    Eq. (1)

where $\lambda_{RL}$ and $\gamma_{RL}$ represent the slope and intercept of the reference line, respectively. Following common practice, we use $\lambda_{RL}$ = 0.528 and $\gamma_{RL}$ = 0 (Luz and Barkan, 2010; Miller et al., 2020).

Mass-independent isotope fractionation during photochemical reactions in the stratosphere produce $CO_2$ that is enriched in $^{17}O$ ($\Delta'^{17}O \gg 0$) (c.f., Boering et al., 2004; Thiemens et al., 1995). Carbon dioxide from high-temperature combustion processes inherits the $^{17}O$-depleted signature of atmospheric $O_2$ ($\Delta'^{17}O \ll 0$) (Horváth et al., 2012). These $^{17}O$-anomalies are largely

reset to near-zero values in the troposphere through mass-dependent isotope exchange between $CO_2$ and water. The $\Delta'^{17}O$ value of tropospheric $CO_2$ therefore provides information on the various natural and anthropogenic $CO_2$ sources and enables quantification of their respective fluxes. Because isotope exchange between $CO_2$ and water occurs predominantly through the terrestrial biosphere, the triple oxygen isotope approach also offers a means of estimating gross primary productivity (Adnew et al., 2025; Barkan and Luz, 2012; Hoag et al., 2005; Hofmann et al., 2017; Horváth et al., 2012; Koren et al., 2019; Liang et

al., 2023; Steur et al., 2024; Thiemens et al., 2014). These applications, however, require analytical methods capable of providing high-precision $\Delta'^{17}O$ measurements ($\leq$ 10 ppm repeatability) rapidly and cost-effectively.

There is a methodological overlap between atmospheric and carbonate research, as both analyse $CO_2$ – in carbonate work, this is commonly obtained via phosphoric acid digestion. A sub-10 ppm repeatability of $\Delta'^{17}O$ measurements in carbonates is also desirable, as such high-precision analyses are necessary to investigate near-equilibrium fractionation processes relevant to

biomineralization and paleoenvironmental research (e.g., Affek et al., 2025; Bajnai et al., 2024; Hare et al., 2025; Kelson et al., 2022).

Stable carbon and oxygen isotope ratios ($\delta^{13}C$ and $\delta^{18}O$) in $CO_2$ are typically measured using isotope ratio mass spectrometry (McCrea, 1950). However, the isotopologue $^{16}C^{13}C^{16}O$ has the same nominal mass as the most abundant $^{17}O$-containing isotopologue $^{16}O^{12}C^{17}O$. This mass interference not only prevents the direct measurement of $\delta^{17}O$ in $CO_2$ using conventional

mass spectrometry but also necessitates corrections in these measurements to account for the $^{17}O$ signal (Brand et al., 2010; Saenger et al., 2021).

To generate $\Delta'^{17}O$ data for $CO_2$, alternative approaches have been developed, most of which produce $O_2$ as the analyte, thereby avoiding mass interference. These techniques include the fluorination of $CO_2$ (Wostbrock et al., 2020), the conversion of $CO_2$ to $H_2O$ followed by the fluorination of $H_2O$ using $CoF_3$ (Brenninkmeijer and Röckmann, 1998; Passey et al., 2014), the

equilibration of $CO_2$ with $O_2$ using a Pt catalyst (Barkan and Luz, 2012; Mahata et al., 2013), and the equilibration of $CO_2$ with

CeO$_2$ followed by fluorination (Hofmann and Pack, 2010; Mahata et al., 2012). High-resolution mass spectrometry has also been used to obtain precise Δ'$^{17}$O data on CO$_2$ (Adnew et al., 2019). However, the above methods are generally too labor-intensive to be practical for routine monitoring of atmospheric CO$_2$.

In recent years, laser spectroscopy has become a viable alternative to mass spectrometry for quantifying rare isotopologues in a range of trace gases, such as CO, CO$_2$, N$_2$O, and CH$_4$ (Becker et al., 1992; Bergamaschi et al., 1994; Lee and Majkowski, 1986; McManus et al., 2006; Mohn et al., 2014; Nataraj et al., 2022; Nelson et al., 2008; Prokhorov et al., 2019; Sakai et al., 2017; Stoltmann et al., 2017; Tuzson et al., 2008; Wang et al., 2020; Yanay et al., 2022; Zhang et al., 2025). Laser spectroscopy determines isotopologue abundances based on the absorption of laser light at specific wavelengths that correspond to the unique vibrational and rotational energy levels of different isotopologues. Consequently, isotopologue ratios obtained through laser spectroscopy are inherently free from mass interference-related biases. For triple oxygen isotope measurements in CO$_2$, two different spectroscopy techniques have been used, cavity ring-down spectroscopy (CRDS) (Chaillot et al., 2025; Stoltmann et al., 2017), and tunable infrared laser direct absorption spectroscopy (TILDAS) (Bajnai et al., 2023; Hare et al., 2022; Perdue et al., 2022; Steur et al., 2021). In this paper, we specifically focus on TILDAS. While some aspects of our discussion relate to the commercially available TILDAS instrument from Aerodyne Research Inc. (Billerica, MA, USA), which also has been used in several recent studies on CO$_2$ triple oxygen isotopes, most apply to the tunable laser absorption spectroscopy in general. Several studies have demonstrated that TILDAS can achieve a repeatability (standard error) of Δ'$^{17}$O measurements of 10 ppm or better, at high speed, while maintaining relatively low analytical costs making it ideal for both air monitoring studies and analyses of carbonate-derived CO$_2$ (Bajnai et al., 2023; Hare et al., 2022; Perdue et al., 2022).

A typical TILDAS setup consists of a laser with a tunable output frequency, a multi-pass cell containing the analyte gas, and a detector (McManus et al., 2006, 2015; Nelson et al., 2008). The Beer-Lambert Law provides the foundation for calculating isotopologue abundances:

$$I_{(\nu)} = I_{o\,(\nu)} \exp\left[ -\frac{N_A}{\ln(10)} \, S_{(T)} \, \phi_{(\nu,P,T)} \, L \, C \right]$$

Eq. (2)

where $I$ and $I_0$ are the intensity of the transmitted and the incident light at frequency ν, respectively. $N_A$ is Avogadro's number (molecules mol$^{-1}$), $S$ denotes the line strength (cm$^{-1}$ (molecule cm$^{-2}$)$^{-1}$), which depends on temperature, $\phi$ the line-shape function (cm), $L$ the absorption path length (cm), and $C$ the concentration of the absorbing species (mol cm$^{-3}$). For CO$_2$ triple oxygen isotope measurements, the abundance of the isotopologues "628" ($^{16}$O$^{12}$C$^{18}$O), "627" ($^{16}$O$^{12}$C$^{17}$O), and "626" ($^{16}$O$^{12}$C$^{16}$O) are quantified based on respective, distinct absorption peaks in the spectral region around 2349 cm$^{-1}$ (Fig. 1). These shorthand notations, commonly used in spectroscopy, identify isotopologues by the second digit of the atoms' atomic masses. The line shape function describes how the absorption of light is distributed around the central wavenumber of a spectral line. It is typically represented by Voigt profile and is derived empirically from the acquired absorption spectra,

measurements of temperatures and pressure as well as spectroscopic fitting parameters retrieved from the high-resolution transmission molecular absorption database (HITRAN) (Gordon et al., 2022).

While variations in the analytical conditions, such as temperature, optical cell pressure, and $CO_2$ concentration have been shown to influence the repeatability of $\Delta'^{17}O$ measurements by TILDAS (Bajnai et al., 2023; Hare et al., 2022, 2025; Perdue et al., 2022; Steur et al., 2021), the extent to which they affect the measured values remain largely unexplored. Understanding the impacts of changing analytical conditions is becoming increasingly important as TILDAS is applied more widely, for example, in long-term atmospheric monitoring studies where analytical conditions are likely to vary (e.g., Steur et al., 2024). In this paper, we examine the sources of analytical uncertainties in two TILDAS setups, located at the University of Göttingen and the University of Cape Town, and evaluate how variations in analytical conditions affect the repeatability of triple oxygen isotope ($\Delta'^{17}O$) measurements of $CO_2$.

## 2 Definitions of spectroscopic isotopologue ratios

TILDAS instruments report scaled isotopologue mole fractions, also referred to as mixing ratios ($\chi'$, commonly expressed as µmol mol$^{-1}$ but Aerodyne Research Inc. instruments report nmol mol$^{-1}$):

$$\chi'_i = C_i \, R \, \frac{T}{P} \, \frac{1}{X_i}$$

Eq. (3)

where $C$ is the concentration (mol m$^{-3}$) of the isotopologue $i$ (e.g., "626", "627", "628"), $R$ is the gas constant (m$^3$ Pa K$^{-1}$ mol$^{-1}$), $T$ and $P$ denote the temperature (K) and pressure (Pa) of the analyte gas, respectively. $X_i$ denotes the isotopologue abundance of the analyzed isotopologue, as defined in the HITRAN database: $X_{626} = 0.98420$, $X_{628} = 0.0039471$, and $X_{627} = 0.000734$ (De Biévre et al., 1984; Gordon et al., 2022). Each isotopologue absorption feature is scaled by the corresponding line strength in the HITRAN database and the reference isotopic abundances. Consequently, the mixing ratios in Eq. (3) are effectively normalized to the total concentration of all isotopologues at natural abundance. Expressing isotopologue abundances as mixing ratios facilitates the calculation of the δ-values (c.f., Griffith et al., 2012; Hare et al., 2022):

$$\delta^{17}O_{meas} = \frac{\chi'_{627,meas}}{\chi'_{626,meas}} - 1$$

Eq. (4)

The scale of these measured δ-values is based on the natural isotopologue abundances used in HITRAN and need to be corrected for instrumental drift. This is done similarly as in isotope ratio mass spectrometry, by repeated alternating measurements of sample ("smp") and a working gas ("wg"), and reporting δ-values relative to the working gas (McKinney et al., 1950):

$$\delta^{17}O_{meas}^{smp/wg} = \frac{\delta^{17}O_{meas}^{smp} + 1}{\delta^{17}O_{meas}^{wg} + 1} - 1$$

Eq. (5)

As in isotope-ratio mass spectrometry, instrumental drift is mitigated by bracketing each sample analysis with measurements of a working gas. This approach relies on the assumption that changes in analytical conditions affect both the sample and the working gas equally, and that sample and working gas have similar molecular composition (i.e. they are matrix-matched).

## 3   Analytical setups

University of Göttingen and the University of Cape Town are both equipped with TILDAS instruments from Aerodyne Research Inc. (Billerica, MA, USA), each coupled with custom-built inlet systems. The design and operation of these systems are described in detail by Bajnai et al. (2023) and Hare et al. (2022). In both laboratories, a single replicate measurement consists of multiple sample cycles bracketed by measurements of a working gas (Eq. 5). However, differences in the inlet systems result in slightly differing analytical procedures.

In TILDAS, mixtures of $CO_2$ and a collision gas (e.g., $CO_2$-free air or pure $N_2$) are used to create optimal spectral line shapes through collisional broadening at pressures between around 30 Torr and 40 Torr (Fig. 1). Matching the $CO_2$ concentration of the sample to that of the working gas, typically within 1 µmol mol$^{-1}$ can be challenging. A key distinction between the setups at Göttingen and Cape Town lies in how the mismatch in $CO_2$ concentrations between the sample and working gas are handled. At Göttingen, the $\chi'_{626}$ of sample and working gas are kept within ±1 µmol mol$^{-1}$ of each other within a replicate (Fig. 2a), and subsequent replicate analyses are also kept within ±1 µmol mol$^{-1}$. This is achieved by mixing pure $CO_2$ analytes, for both reference and sample, with a collision gas to a predefined target prior to each respective measurement cycle. In contrast, at Cape Town, the reference gas is pre-mixed and taken from a 50 L high pressure cylinder of 421 µmol mol$^{-1}$ $CO_2$ in Nitrogen 5.0, resulting in identical $\chi'_{626}$ values of the working gas across replicate analyses within 0.1 µmol mol$^{-1}$ (1σ). The $\chi'_{626}$ of the sample, however, varies from sample to sample, depending on the amount of sample gas available for analysis (Fig. 2b).

Another difference between the two laboratories lies in the temperature control of the TILDAS. The optical cell and some electronic components of the Aerodyne TILDAS instruments are thermally stabilized by a recirculating liquid chiller. To prevent large and rapid room temperature fluctuations that the chiller cannot compensate for, the Göttingen instrument is housed in an insulated box with PID temperature control, maintaining the ambient temperature stable within ±0.01 K. In Cape Town, the instrument is not custom-insulated but is housed in a laboratory with an exceptionally stable temperature, varying by only ±0.1 K over the course of a one-hour measurement.

## 4   Dependence of the measured $\Delta'^{17}O$ values on the analytical conditions

### 4.1   Concentration dependence due to scale-offset

The effect of mismatched $CO_2$ concentrations between sample and working gas have been shown to substantially affect the measured $\Delta'^{17}O$ values and requires correction (Bajnai et al., 2023; Hare et al., 2022, 2025; Steur et al., 2021, 2024). According

to Eqs. 2 and 3, the measured $\chi'$ values are a function of the area under the corresponding spectral peaks [e.g., $\chi'_{626} \propto A(626)$]. The size of the absorption peaks varies with $pCO_2$. Under perfectly ideal instrumental conditions, the observed ratio of the peak areas would remain constant even with a change in $pCO_2$ [$\delta^{17}O_{meas} \propto a \times A(627)/A(626) + b$, with $a=0$ and $b=0$]. However, under real world conditions changes in peak areas are not perfectly retrieved introducing concentration dependence of the ratios of the peak areas, which leads to a concentration dependence of the measured $\delta$-values ($\delta^{17}O_{meas} \propto a \ A_{627}/A_{626} + b$, with $a \neq 0$ and $b \neq 0$). The underlying causes of this effect are related to systematic errors in the measurement of mole fractions and likely include nonlinearities related to spectral retrievals and the infrared detector response.

Deviations in the measured $\chi'$ values from the "true" mole fractions of the analyte gas can be expressed as (c.f., Griffith et al., 2012; Hare et al., 2022):

$$\chi'_{627,meas} = a_{627} \times \chi'_{627,true} + b_{627}$$

Eq. (6)

where $a_{627}$ and $b_{627}$ are scaling factors which relate the measured isotopologue mole fractions to the "true" isotopologue mole fractions of the analyte gas, either sample or working gas. Similar equations can be written for $\chi'_{626}$ and $\chi'_{628}$.

Using Eq. (6), we can describe a more accurate form of Eq. (4), adjusted for the offsets:

$$\delta^{17}O_{meas} = \frac{a_{627} \times \chi'_{627,true} + b_{627}}{a_{626} \times \chi'_{626,true} + b_{626}} - 1$$

Eq. (7)

From Eqs (6) and (7), the "true" $\delta$-values of sample or working gas can be expressed as follows (see Eq. 4 in Hare et al., (2022)):

$$\delta^{17}O_{true} = \frac{\chi'_{626,meas}}{\frac{a_{627}}{a_{626}} \left(\chi'_{626,meas} - b_{626}\right)} \left( \delta^{17}O_{meas} + \frac{\frac{a_{627}}{a_{626}} \times b_{626} - b_{627}}{\chi'_{626,meas}} - \frac{a_{627}}{a_{626}} + 1 \right)$$

Eq. (8)

Equation (8) implies that all $\delta$-values, and consequently $\Delta'^{17}O$ values, measured by spectroscopy depend on the measured mole fraction of the most abundant $CO_2$ isotopologue, $\chi'_{626}$. Therefore, variations in $\chi'_{626}$ across successive measurement cycles within a replicate or across individual replicate analyses affect measurement repeatability.

To correct for $pCO_2$ mismatch, the "true", scale-offset-corrected $\delta$-values need to be determined. In practical terms, this means that the constants $a$ and $b$ in Eq. (7) need to be known. Instead of correcting the $\delta$-values values, it is convenient to perform the correction on the measured $\Delta'^{17}O$ values directly. By combining Eqs (1, 5, 7, 8) we can derive a correction scheme for "true" $\Delta'^{17}O_{smp/wg}$ values relative to the working gas used for bracketing (see Appendix A):

$$\Delta'^{17}O_{true}^{smp/wg} \simeq \Delta'^{17}O_{meas}^{smp/wg} - m \times \left( \chi'^{smp}_{626,meas} - \chi'^{wg}_{626,meas} \right)$$

Eq. (9)

To determine the correction slope $m$ empirically, we carried out a series of experiments. At Cape Town, 10 mg of IAEA-603 was reacted in 100% phosphoric acid at 70 °C and then diluted with varying amounts of $N_2$. This resulted in $\chi'^{smp}_{626}$ ranging from 360 µmol mol$^{-1}$ to 435 µmol mol$^{-1}$, while the $\chi'^{wg}_{626}$ was held constant at 421 µmol mol$^{-1}$. A similar experiment was done by diluting the working reference gas. A fit to this data — measured within the same analytical session (10–12 March 2025) — yields a slope of $m = -6$ ppm per µmol mol$^{-1}$ mismatch ($\chi'^{smp}_{626} - \chi'^{wg}_{626}$), both for IAEA-603 and for the zero-enrichment measurements (Fig. 3). A second set of experiments conducted during a separate analytical session (21 November – 10 December 2024) yielded a slightly different slope for IAEA-603 of $m = -7$ ppm per µmol mol$^{-1}$ mismatch (Fig. 3).

At Göttingen, we used the internal light $CO_2$ and heavy $CO_2$ reference gases for similar dilution experiments. For the 2023 analytical session (15–19 March 2023), both analytes yielded a slope of $m = 6$ ppm per µmol mol$^{-1}$ mismatch. After the laser was replaced in 2025, the same dilution experiments (30 June – 14 July 2025) produced a slope of $m = 15$ ppm per µmol mol$^{-1}$ mismatch (Fig. 3).

The observation that gases with different isotope compositions ($\Delta\delta^{18}O$ relative to the working gas ranging from -28‰ to +48‰) yield identical correction slopes $m$ within the same analytical sessions suggests that $m$ is largely independent of the isotopic composition of the sample analyte.

The opposite sign of $m$ observed in Göttingen and Cape Town (Fig. 3) can be explained by differences in the sign of the expression $a_{626}/b_{626}$. This, together with the variation in the correction slopes across the analytical sessions, indicates that $m$, that is ultimately $a_{626}$ and $b_{626}$, are instrument dependent and may vary in response to changing instrumental parameters, such as the laser tuning rate, the instrument purging rate, etc.

We apply the correction described by Eq. 9 to address the mismatch between the $\chi'_{626}$ of the sample and the working gas in the Cape Town dataset. Before correction, the external 1 standard deviation of the Cape Town NBS-18 and IAEA-603 $\Delta'^{17}O$ values is 110 ppm and 98 ppm, respectively, over the entire measurement period (Fig. 4a). After applying apply a scale-offset correction to the entire dataset, these values improve significantly to 32 ppm and 38 ppm (Fig. 4b). Applying a session-specific correction (i.e. with different $m$ fitted to each analytical session), there is further improvement to 20 ppm for both NBS-18 and IAEA-603 (see details in Hare et al., 2025).

The correction scheme outlined above is only valid for small differences in $\chi'_{626}$ between sample and working gas. We estimate a ±1% error in the linear correction slope $m$, if $\chi'^{smp}_{626}$ is within approximately 14% of $\chi'^{wg}_{626}$, i.e. ±58 ppm for a typical $\chi'^{wg}_{626}$ value of 420 ppm (see Appendix A). That is, if $m = 6$ ppm per µmol mol$^{-1}$ mismatch, as observed at both laboratories, the error arising from the correction while correcting for a 58 ppm mismatch is about 3 ppm.

We briefly note that other techniques, e.g., CRDS, use pure $CO_2$ gas as analyte instead of gas mixtures (c.f., Chaillot et al., 2025). According to Eq. (9), pure $CO_2$ implies that $\chi'^{smp}_{626} - \chi'^{wg}_{626} \simeq 0$, eliminating the need for a scale-offset correction.

## 4.2 Temperature dependence

Temperature variability could affect a laser spectroscopy system in multiple ways. Because the line shape function is empirically derived from the absorption spectra, the calculated concentrations are inherently subject to some inaccuracy (Eq. 2). The empirical fit may not fully capture subtle variations in thermal or collisional peak broadening, resulting in residual temperature and pressure dependencies in the measured concentrations (Becker et al., 1992; Bergamaschi et al., 1994; Nelson et al., 2008). These biases can affect each spectral peak, and therefore each measured mixing ratio, differently, leading to uncorrelated changes in the measured $\delta$-values, and, ultimately, variations in the $\Delta'^{17}O$ values.

The temperature sensitivity of the electronics components could also affect the measured isotope ratios. Temperature variations could affect the sensors, such as the infrared detector, or pressure sensor. The output power and wavelength of a laser are affected by temperature in two main ways: through the temperature of the laser itself and through the temperature sensitivity of the current driver, which influences the laser current (e.g., Hare et al., 2022; Nelson et al., 2008; Wavelength Electronics, 2020).

Figure 5 shows a 10-hour measurement of a $CO_2$-in-air mixture conducted at the University of Göttingen using TILDAS (cf. Fig. 3 in Bajnai et al. (2023)). This uninsulated prototype setup was placed in an air-conditioned laboratory without a strict temperature control. Variations in room temperature are clearly reflected in both the electronics (by 2 K) and coolant temperatures (by 0.07 K). The cell temperature also closely tracks the fluctuations in the room temperature, albeit with a ca. 20 min delay and a reduced amplitude (by 0.07 K) due to the large metal mass of the cell and its active temperature control.

The temperature variations are broadly reflected in the measured concentrations of the "626", "627", and "628" isotopologues. However, these changes are not identically mirrored across the three isotopologue concentrations (Fig. 5b), leading to uncorrelated variations in the $\delta$-values (Fig. 5c) and resulting in a drift of about 1500 ppm in the measured $\Delta'^{17}O$ values (Fig. 5d).

The alignment of the isotope ratio variations with the electronics temperature, rather than with the cell temperature, suggests that the temperature sensitivity of the $\Delta'^{17}O$ measurements, in this case, is primarily linked to the temperature sensitivity of the electronic components rather than to the empirical fitting of the line shape function.

## 4.3 Gas purity

A major advantage of optical isotope ratio measurement techniques over mass spectrometry is that they are inherently free from isobaric interferences. However, optical measurements can still be affected by contaminants, particularly gases that exhibit overlapping absorption features or contribute to peak broadening.

In the spectral window used for triple oxygen isotope analyses of $CO_2$, the three relevant $CO_2$ isotopologues produce distinct absorption peaks (Fig. 1), with no overlapping signals from water or other common contaminant gases such as $N_2O$, $NO_2$, CO,

or $SO_2$ (c.f., Gordon et al., 2022). For completeness, we note that this advantage does not extend to $\Delta_{47}$ clumped isotope analyses, where $N_2O$ exhibit peaks within the relevant spectral window (Yanay et al., 2025). In such cases, a robust sample purification is required to remove interfering species.

The absence of direct spectral overlap does not imply that analyte purity is irrelevant. Variations in the gas composition — particularly the presence of water vapor — can lead to peak broadening effects, as well as isotopic exchange between $CO_2$ and water. Corresponding changes in the measured molar concentrations, in turn, contribute to variability in isotope ratios across replicate analyses. This effect has been well documented in atmospheric $CO_2$ monitoring studies (e.g., Paul et al., 2020; Tuzson et al., 2008), and may also be relevant for analyses of $CO_2$ produced by the acid digestion of carbonates. Although the water

content of the released $CO_2$ analyte is typically low, it is not negligible and may vary depending on acid temperature and density (c.f. Wacker et al., 2013).

An additional source of gas-purity-related error arises when the gas matrices of the sample and the working gas differ. For example, the spectroscopic fitting parameters used by the Aerodyne Research Inc. TDLWintel software are based on HITRAN data, which assume peak broadening due to collisions in a $CO_2$-free air matrix. If the actual matrix deviates substantially from

245 this assumption, spectral fits may degrade. This was observed by Bajnai et al. (2023), who noted to encounter fitting issues when using argon as a bathing gas without adjusting the fit parameters.

Changes in the gas matrix can also influence the scaling factors $a$ and $b$ in Eqs. (6–9). Gas matrix effects may become a significant source of uncertainty in air monitoring studies, particularly when the gas matrix of the working gas (e.g., $CO_2$-in-$N_2$) differs from that of the sample ($CO_2$-in-air with variable argon concentrations), since the resulting scale-offsets can differ

and be difficult to correct. Matching the matrices of the working gas to that of the sample analyte as closely as possible helps prevent detrimental effects arising from variable scale-offsets.

A similar gas-matrix–related issue can arise when the matrix of the reference gas used for standardization differs from that of the sample gas. For example, consider a pre-mixed $CO_2$-in-$N_2$ reference gas used to correct measurements of $CO_2$-in-air. In such cases, even if the analytical conditions remain completely stable, gas matrix effects may be carried into the correction,

ultimately biasing the data.

## 5    Internal error and long-term drift in the Göttingen and the Cape Town datasets

### 5.1    Internal error

Variations in the analytical conditions can introduce instrumental drift. Instrumental drift is mitigated by bracketing each sample analysis with measurements of a working gas (McKinney et al., 1950). This approach relies on the assumption that

changes in analytical conditions affect both the sample and the working gas equally. It also assumes that the changeover between cycles is fast enough for variations in analytical parameters to be approximated by a linear trend between two working

gas cycles. The internal error of a single replicate analysis, i.e. the repeatability of approximately 10 sample cycles within a bracketing measurement, primarily depends on how constant the measurement conditions remain between cycles.

We assess the variability and stability of key analytical parameters — cell temperature, electronics temperature, cell pressure, and the mixing ratio of the most abundant $CO_2$ isotopologue ($\chi'_{626}$) — using four indicators. First, we determined the mean value of each parameter separately for sample and reference cycles within each replicate. Second, we used the range of these parameter means across replicates as a measure of long-term drift in analytical conditions. Third, to quantify systematic deviations between sample and reference measurements, we calculated, for each sample cycle, the difference between its mean value and the mean of the reference cycles immediately before and after. The average of these differences (sample minus reference) is referred to as the mismatch parameter. A mismatch may not decrease the repeatability if, for example it is constant over the cycles of a replicate analysis. However, variations in mismatch values, both within a single replicate and across replicates, affects measurement repeatability. To assess the stability of the analytical conditions within a single replicate, we calculated the standard deviation of the mismatch values within a replicate.

Figure 6 shows the relationship between the stability of the mismatch in the mixing ratio of the most abundant $CO_2$ isotopologue ($\chi'_{626}$), cell pressure, cell temperature, electronics temperature, and the internal error of individual replicate measurements (68% confidence interval of the approximately 10 sample cycles bracketed by working gas analyses). At the University of Göttingen, the $\chi'_{626}$ mismatch across the cycles of a single replicate remains stable within ±1 µmol mol⁻¹ (Fig. 6a). The cell pressure mismatch stays stable within ±0.8 Pa (approximately ±6 mTorr; Fig. 6c), and the cell temperature mismatch within ±1 mK (Fig. 6e). The electronics temperature stays stable within ±0.10 K (Fig. 6g). None of these parameters show a statistically significant correlation with the observed internal error, suggesting that these variables are kept stable enough during a replicate measurement to avoid influencing the results.

At the University of Cape Town, the across the cycles of a single replicate the mismatch in $\chi'_{626}$, cell pressure, cell temperature and electronics temperature remain stable better than ±12 µmol mol⁻¹ (Fig. 6b), 34.7 Pa (approximately ±260 mTorr; Fig. 6d), ±3 mK (Fig. 6f), and ±0.14 K (Fig. 6h), respectively. Weak correlations (up to $R^2 = 0.2$) between the mismatch stabilities and the internal error appear to be driven by outlier points. Overall, the lack of meaningful correlation between the stability of the mismatch parameters and the internal error of the $\Delta'^{17}O$ measurements suggests that the stability of the analytical parameters is sufficiently stable in both laboratories to prevent any systematic effect on internal error.

## 5.2 Long-term drift in the analytical parameters

The external repeatability — i.e. the consistency of results across multiple independent replicate analyses — primarily depends on the stability of measurement conditions. Rather than focusing on the drift of individual reference materials, we examine the difference in measured $\Delta'^{17}O$ values between two reference materials. We refer to this difference as scale compression, adopting terminology similar to that used in mass spectrometry. This approach offers two key advantages: it simplifies the

identification of trends in the data and, more importantly, directly relates to the most common isotope standardization strategy, which relies on two-point calibration using reference materials measured within a defined period. In this context, drift in the scale compression within a measurement period directly affects the accuracy of the final $\Delta'^{17}O$ values.

Figures 7 and 8 illustrate the long-term drift in the analytical parameters and the measured $\Delta'^{17}O$ values of two reference materials and the corresponding scale compression in the Göttingen and Cape Town datasets. A LOESS fit is used to approximate temporal trends in each variable. To assess which variables most strongly influence the drift in the scale compression, we performed a multiple linear regression analysis, relating various combinations of analytical parameters to the scale compression (Fig. 9). Note that in the case of the Cape Town dataset $\chi'_{626}$ is not considered, as $\Delta'^{17}O$ values have already been corrected to a constant $\chi'_{626}$ value of 421 µmol mol$^{-1}$ (c.f., Fig. 4). The $R^2$ values of the multiple linear regression analysis indicate that in Göttingen, variations in electronics temperature is the primary driver of the scale compression drift, while cell temperature, cell pressure, and $\chi'_{626}$ exert smaller yet still considerable effects (Fig. 9a). Considering only variations in $\chi'_{626}$ in the regression model does not explain the long-term drift in the scale compression. This result is consistent with expectations: given the $\chi'_{626}$ sensitivity of $\Delta'^{17}O$ of 6 ppm per µmol mol$^{-1}$, the observed ca. 2.5 µmol mol$^{-1}$ variation in $\chi'_{626}$ would lead to a maximum drift of about 15 ppm, that is comparable to the 1σ repeatability of measurements under ideal conditions. By contrast, the observed overall ~60 Pa variation in cell pressure and ~0.1 K variation in cell temperature produce more substantial effects on the scale compression. Variations in cell as well as electronics temperature and cell pressure are the dominant contributors to the scale compression drift in Cape Town (Fig 9b). Figures 7f and 8e illustrate the combined parameter drift resulting from the analytical parameters combinations yielding the highest $R^2$ values in Fig. 9.

Figure 7a illustrates that the magnitude and direction of the drift in the $\Delta'^{17}O$ values of the two standards used in the Göttingen laboratory are not identical. The $\delta^{18}O$ values of these two standards differ by ca. 80‰. As discussed above and shown in Fig. 5, the mixing ratios of the three $CO_2$ isotopologues respond in an uncorrelated fashion to variations in analytical conditions related to systematic errors in the measurement of mole fractions. It follows that the magnitude of the drift in the measured $\delta$ and $\Delta'^{17}O$ values in response to changing analytical conditions, particularly temperature, depends on the isotopic composition of the analyte.

## 6   Summary and recommendations

Variations in analytical conditions — such as cell temperature, electronics temperature, cell pressure, and the $pCO_2$ of the analytes — can influence the measured $\Delta'^{17}O$ values. For instance, the sensitivity of $\Delta'^{17}O$ to changes in analyte $pCO_2$ is on the order of 10 ppm per µmol mol$^{-1}$, depending on the instrumentation. This source of bias is particularly relevant for atmospheric monitoring studies, as atmospheric $CO_2$ concentrations can fluctuate by more than 100 µmol mol$^{-1}$ over the course of a day, potentially causing a variability of 1000 ppm in the measured $\Delta'^{17}O$ values that needs to be corrected.

In addition to $p\text{CO}_2$, instability in cell temperature, electronics temperature, and cell pressure can compromise measurement repeatability (internal error). Data from the University of Göttingen and the University of Cape Town show that maintaining

cell temperature and pressure stability within $\pm 1$ mK and $\pm 10$ Pa, respectively, across the cycles of a replicate measurement is sufficient to avoid any resolvable impact on the internal error beyond $\pm 10$ ppm on $\Delta'^{17}O$. The temperature sensitivity of the instrumental setup depends on the relative speed of the changeover measurements to the speed of the variation in the ambient temperature. To minimize the influence of ambient temperature fluctuations, TILDAS systems are typically placed in thermally insulated enclosures or operated in climate-controlled laboratories.

Long-term drift in the analytical conditions — such as a gradual changes in electronics temperature, cell temperature and pressure, and mixing ratios over the course several weeks — can introduce drift in the measured $\Delta'^{17}O$ values. As the magnitude of the drift in the measured $\Delta'^{17}O$ values seemingly depend on the isotope composition of the analyte, the suitability of a polynomial fit based on reference materials for correcting drift in samples needs to be confirmed. If left unrecognized, these drifts may be misinterpreted as genuine seasonal variations in $\Delta'^{17}O$ data. Continuous monitoring and reporting of the

analytical conditions, along with periodical recalibration of temperature and pressure sensors, are therefore essential to ensure data integrity over extended timescales.

Finally, we demonstrated that changes in analytical conditions can impact the instrument's scaling, that is, the measured difference in $\Delta'^{17}O$ between two reference materials used for data correction. To ensure accurate data correction, subsequent measurements should be grouped into measurements periods under which variations in the analytical conditions do not

introduce a resolvable effect on the measured oxygen isotope values. The duration of the measurement periods and the required number of reference measurements necessary for a precise correction depends on the specific analytical setup.

## 7  Appendix A

The following text outlines the derivation of Eq. (9). Equation (8) describes the concentration dependence of the $\delta$-values of sample and working gas. Substituting $a_{627}/a_{626}$ with $A_{627}$, and $a_{628}/a_{626}$ with $A_{628}$, following the nomenclature of Hare et al

(2022), yields:

$$\delta^{17}O_{\text{true}} = \frac{\chi'_{626,\text{meas}}}{A_{627}\left(\chi'_{626,\text{meas}} - b_{626}\right)}\left(\delta^{17}O_{\text{meas}} + \frac{A_{627} \times b_{626} - b_{627}}{\chi'_{626,\text{meas}}} - A_{627} + 1\right)$$

Eq. (A1)

The fitted coefficients in Hare et al. (2022) provide indicative numerical values of $A_{627} = 0.674$, $A_{628} = 0.632$, $b_{626} = -168$, $b_{627} = -1974$, $b_{628} = -4782$, when $\chi'_{626}$ is expressed in nmol mol$^{-1}$.

Assuming that $|\chi'_{626}| \gg |b_{626}|$, $|b_{626}| \gg |A_i|$, and $|b_i| \approx |b_{626}|$, with $i$=627,628, we can approximate Eq. (A1) as:

$$\delta^{17}O_{true} \simeq \frac{\chi'_{626,meas}}{A_{627}(\chi'_{626,meas} - b_{626})}\delta^{17}O_{meas}$$

Eq. (A2)

This approximation preserves the concentration dependence. Evaluating the prefactor with the fitted coefficients above, yields a multiplier of approximately 1.48. This multiplier will be different on each system, depending on the value of $\chi'_{626}$, as well as $b_{626}$ and $A_{627}$ (or $A_{628}$ for $\delta^{18}O$).

From Eqs. (1) and (5) we have:

$$\Delta'^{17}O_{true}^{smp/wg} = \Delta'^{17}O_{true}^{smp} - \Delta'^{17}O_{true}^{wg}$$

Eq. (A3)


Under the assumptions outlined above, we can approximate that $\ln(\delta^{17}O_{true} + 1) \approx \ln[\delta^{17}O_{true} + \chi'_{626} / (A_{627} (\chi'_{626} - b_{626})]$. With Eqs. (1, 5, 7), and a separate form of Eq. (A2) for sample and working gas, Eq. (A3) can be evaluated as:

$$\Delta'^{17}O_{true}^{smp/wg} = \Delta'^{17}O_{meas}^{smp/wg} + (1 - \lambda_{RL}) \ln \left[ \frac{\chi'^{smp}_{626,meas}(\chi'^{wg}_{626,meas} - b_{626})}{\chi'^{wg}_{626,meas}(\chi'^{smp}_{626,meas} - b_{626})} \right]$$

Eq. (A4)

Under the assumption that $|\chi'_{626}| >> |b_{626}|$, a Taylor-series expansion of Eq. (A4) yields the following hyperbolic correction

equation, to first order:

$$\Delta'^{17}O_{smp/wg}^{true} \simeq \Delta'^{17}O_{smp/wg}^{meas} + (1 - \lambda_{RL})b_{626}\left[ \frac{1}{\chi'^{smp}_{626,meas}} - \frac{1}{\chi'^{wg}_{626,meas}} \right]$$

Eq. (A5)

From here, it is convenient to linearize the hyperbola Eq. (A5) for large values (i.e. 420 000 nmol mol$^{-1}$) of $\chi'^{wg}_{626}$ and small differences between $\chi'^{wg}_{626}$ and $\chi'^{smp}_{626}$:

$$\Delta'^{17}O_{smp/wg}^{true} \simeq \Delta'^{17}O_{smp/wg}^{meas} - (1 - \lambda_{RL})b_{626}/(\chi'^{wg}_{626,meas})^2 \times (\chi'^{smp}_{626,meas} - \chi'^{wg}_{626,meas})$$

Eq. (A6)

From Eq. (A6) we get Eq. (9), with the correction slope $m$ as $(1 - \lambda_{RL}) * b_{626} / (\chi'^{wg}_{626})^2$,

The condition for a maximum 1% error in the linear approximation of the hyperbola (Eq. A6) is $(\chi'^{smp}_{626} - \chi'^{wg}_{626})/(2(\chi'^{wg}_{626})^2)$ $\leq 0.01$, which implies that $|\chi'^{smp}_{626} - \chi'^{wg}_{626}| \lesssim 0.14 \chi'^{wg}_{626}$. In other words, the concentration of the sample should be within 14% of the working gas, to avoid errors of greater that 1% in the correction. For example, if $m$ is 6 ppm per µmol mol$^{-1}$, then at a mismatch of 58 µmol mol$^{-1}$, there will be an absolute error in corrected, "true" $\Delta'^{17}O_{smp/wg}$ of approximately 3 ppm due 370 to a deviation in linearity.

## 8   Code and data availability

All data and codes used in this manuscript are deposited at GitHub (https://github.com/davidbajnai/TILDAS_drift) and Zenodo (https://doi.org/10.5281/zenodo.15742110) (Bajnai and Hare, 2025). The supplementary files include Table S1 (long-term

replicate-level measurement data from the University of Göttingen), Table S2 (long-term replicate-level measurement data from the University of Cape Town), Table S3 (concentration experiments from the University of Cape Town), and Table S4 (concentration experiments from the University of Göttingen).

## 9   Author contribution

DB: Conceptualization, Formal analyses, Investigation Methodology, Writing – original draft, Visualization

VJH.: Conceptualization, Formal analyses, Investigation Methodology, Writing – original draft

## 10  Competing interests

The authors declare that they have no conflict of interest.

## 11  Acknowledgements

David Bajnai gratefully acknowledges Prof. Andreas Pack, head of the stable isotope laboratory at the University of Göttingen, for his generous support and for providing the infrastructure that made this research possible. He also thanks Dennis Kohl, Thierry Wasselin, and Tommaso Di Rocco for their technical assistance. Vincent Hare gratefully acknowledges the laboratory assistance of Drake Yarian and Anna Kudriavtseva at the University of Cape Town.

## 12  Financial support

This work was supported by the Department of Science and Innovation (Republic of South Africa), through funding from the Biogeochemistry Research Infrastructure Platform (BIOGRIP).

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

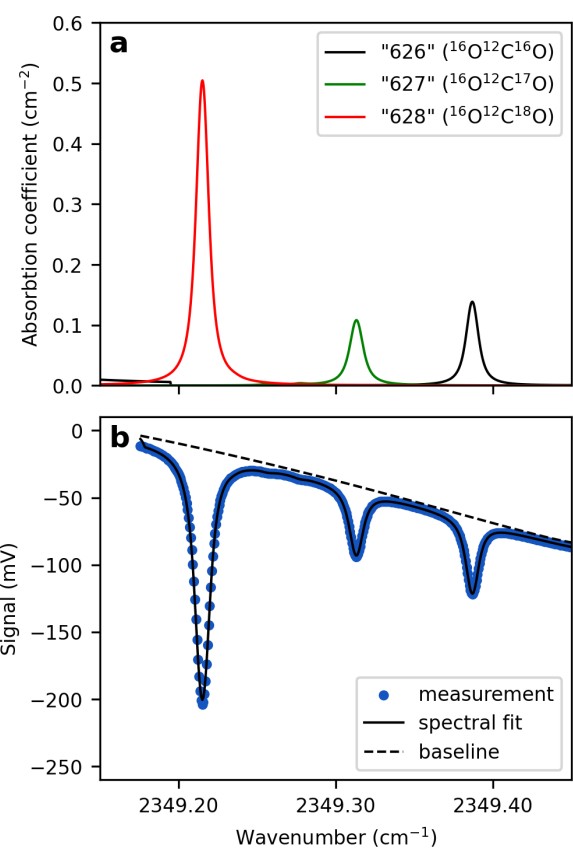

**Figure 1. Modeled and measured CO₂ absorption spectra for Δ'¹⁷O analysis**

a) Absorption coefficients of the three $CO_2$ isotopologues used for triple oxygen isotope measurements, retrieved from the HITRAN database. The broadening coefficients reflect the experimental conditions below: $P = 41.335$ Torr, $T = 297.6$ K, and $p$CO₂ = 420 μmol mol⁻¹. b) Measured spectrum of a CO₂-in-air gas mixture obtained using TILDAS. Green dots represent individual data points, each corresponding to an average of 1538 spectra acquired at a rate of one spectrum per second. The TDLWintel software fits the spectrum to the data (blue line), accounting for several parameters, including the analyte's temperature and pressure, as well as spectroscopic information from the HITRAN database. Diluting a pure $CO_2$ analyte with a collision gas broadens the absorption peaks, allowing each peak to be represented by more data points.

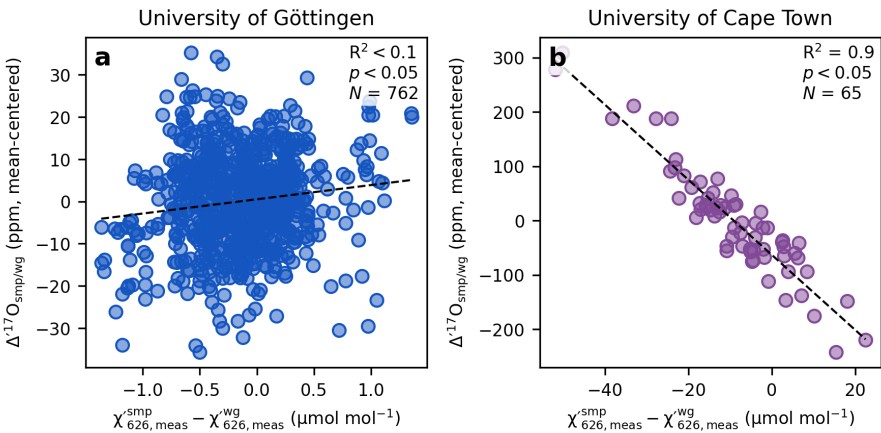

**Figure 2. The effect of $p\mathrm{CO_2}$ mismatch between sample and working gas on the measured $\Delta'^{17}\mathrm{O}$ values.**

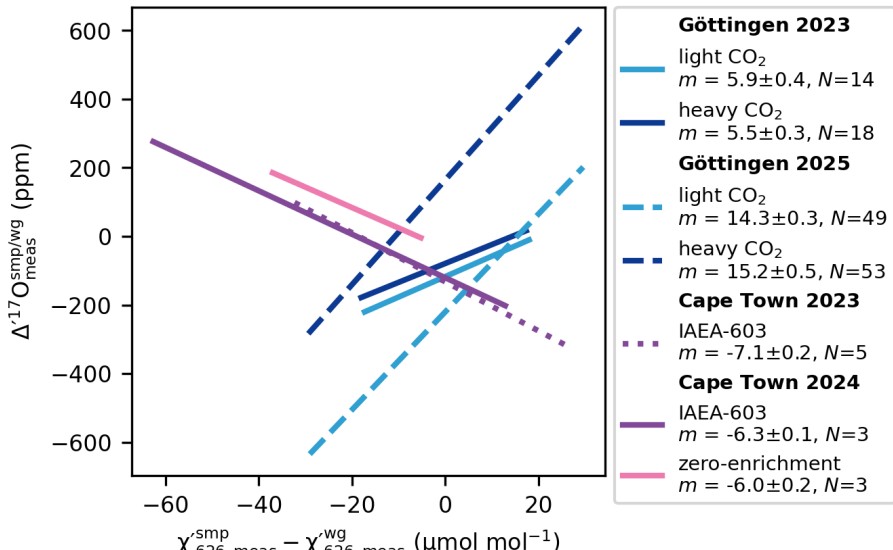

**Figure 3. Concentration dependence of $\Delta'^{17}O$**

Linear trendlines showing the relationship between mismatches in the measured $\chi'_{626}$ mixing ratios of sample and working gas and corresponding $\Delta'^{17}O$ values. The figure also highlights that the slopes vary not only between instruments but also among analytical sessions of the same instrument. The corresponding concentration dependence of the measured $\delta^{18}O$ values is smaller than $\pm0.03‰$ per µmol mol$^{-1}$ mismatch in $\chi'_{626}$.

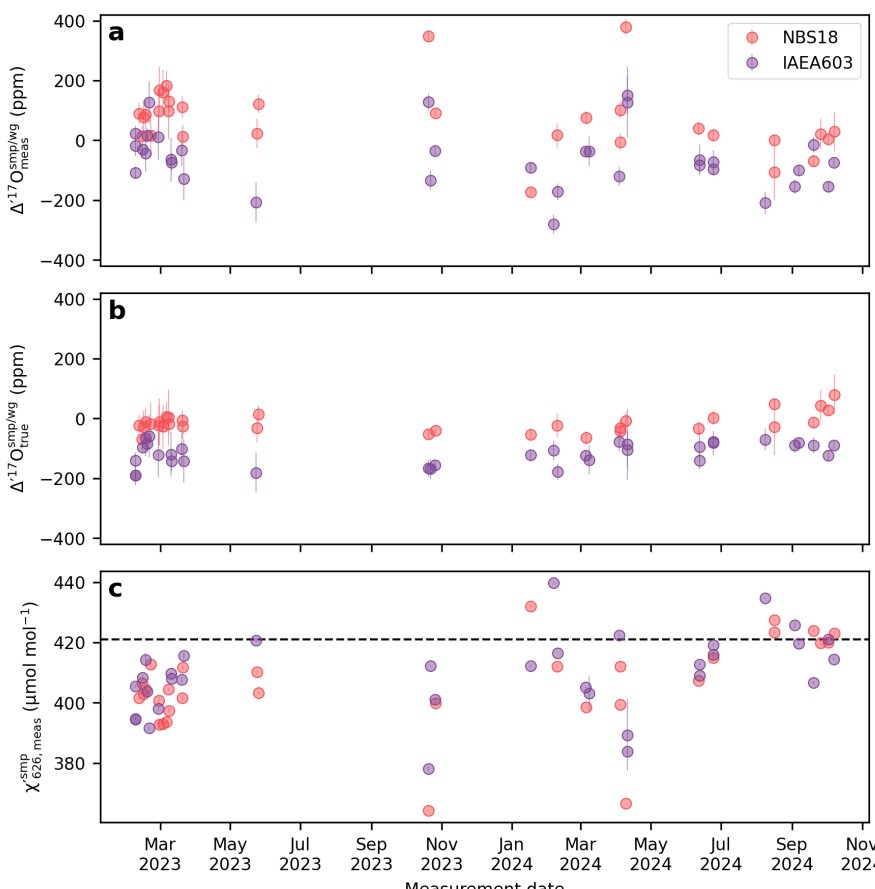

**Figure 4. The effect of scale-offset correction of the Cape Town Δ'¹⁷O data.**

a) Measured $\Delta'^{17}O$ values, b) "true" $\Delta'^{17}O$ values after scale-offset correction, c) Average $\chi'_{626}$ values of the sample gas within a replicate analysis. For the Cape Town setup the $\chi'_{626}$ value of the working gas is a constant 421 μmol mol⁻¹, and within a replicate measurement the mismatch between subsequent $\chi'_{626}$ values stay within ±1 μmol mol⁻¹.

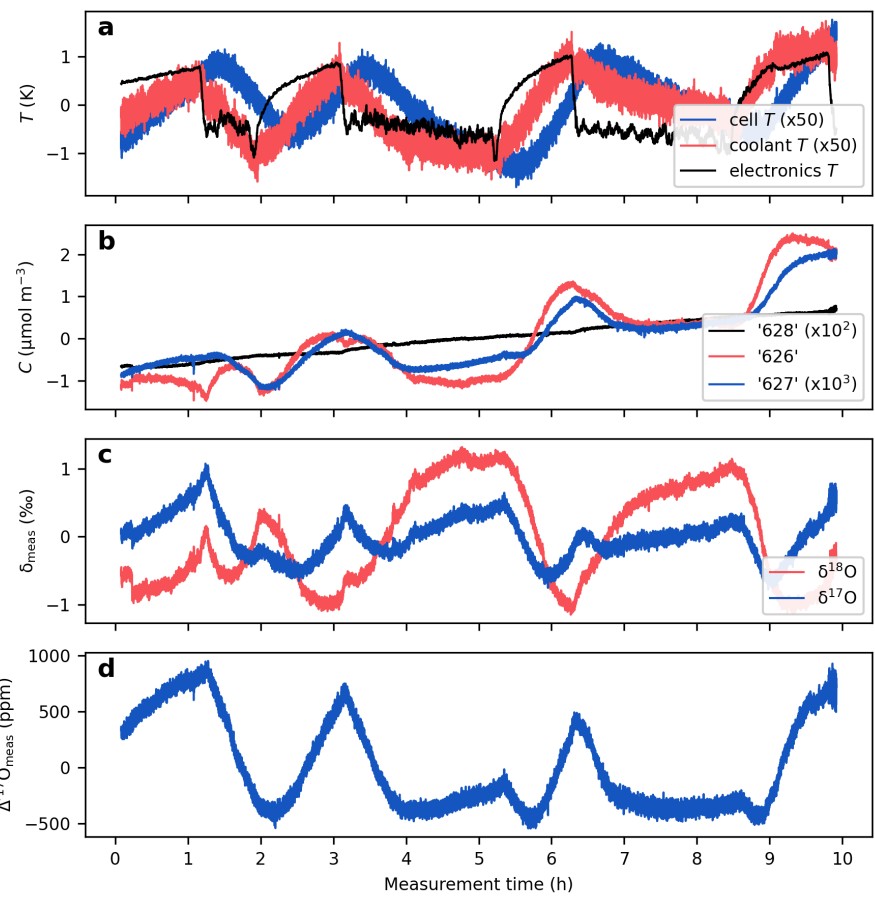

**Figure 5. Temperature effects on the measured isotope ratios.**

The figure presents a continuous measurement of a $CO_2$-in-air mixture at 420 µmol mol$^{-1}$. a) Instrumental temperatures. Coolant temperature refers to the temperature of the circulating liquid entering the instrument. Electronics temperature represents the temperature measured adjacent to the electronics boards and the laser current driver. Note the approximately 20-minute lag between the cell temperature and the electronics and coolant temperatures, which result from the thermal inertia of the metal cell. b) Isotopologue concentrations calculated from mixing ratios, cell temperature and cell pressure (Eq. 3). c) δ-values; d) Δ'$^{17}$O values. All data are mean centered to highlight relative variations.

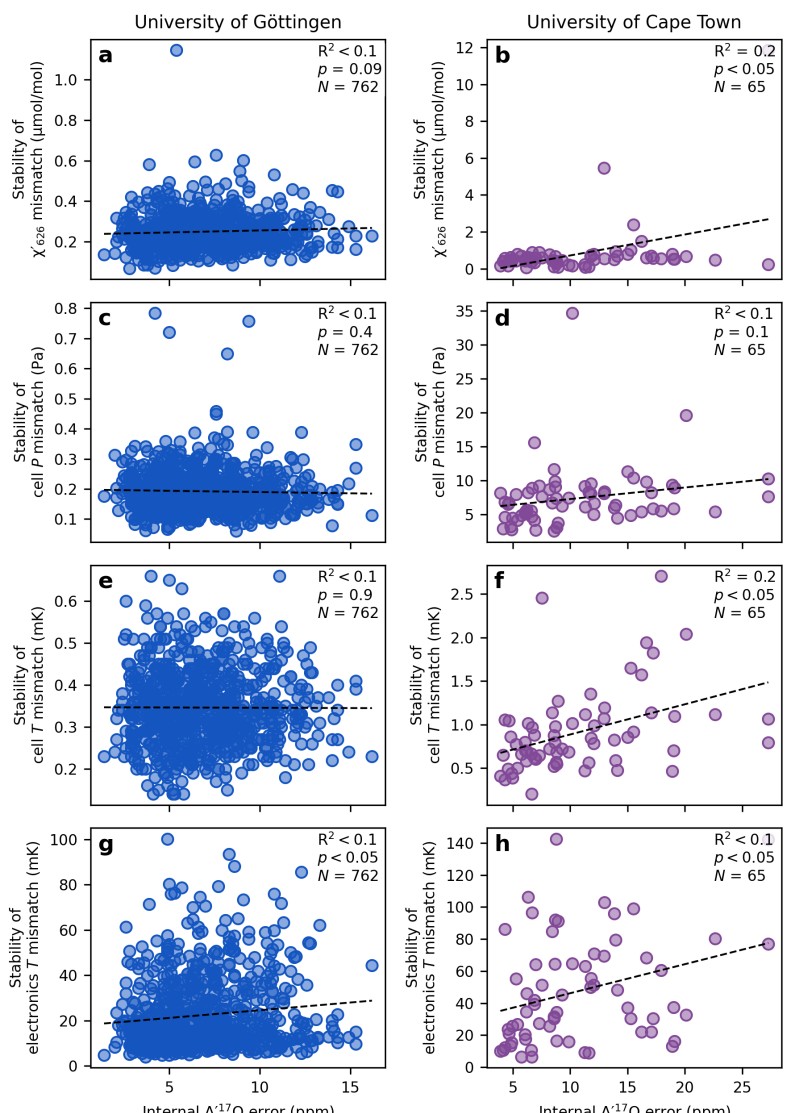

**Figure 6. Effect of analytical variability on the internal error of a replicate Δ'¹⁷O measurement.**

The standard deviation of the mismatch parameters serves as a measure of variability in analytical conditions across cycles of individual replicate analyses. The internal error is reported as the 68% confidence interval of the calculated Δ'¹⁷O values from approximately 10 sample cycles. The black dashed line and the corresponding correlation coefficients indicate the relationship between parameter stability and internal repeatability. In both laboratory setups, the analytical conditions are generally stable enough that they do not have a detectable effect on the internal reproducibility of the measurements.

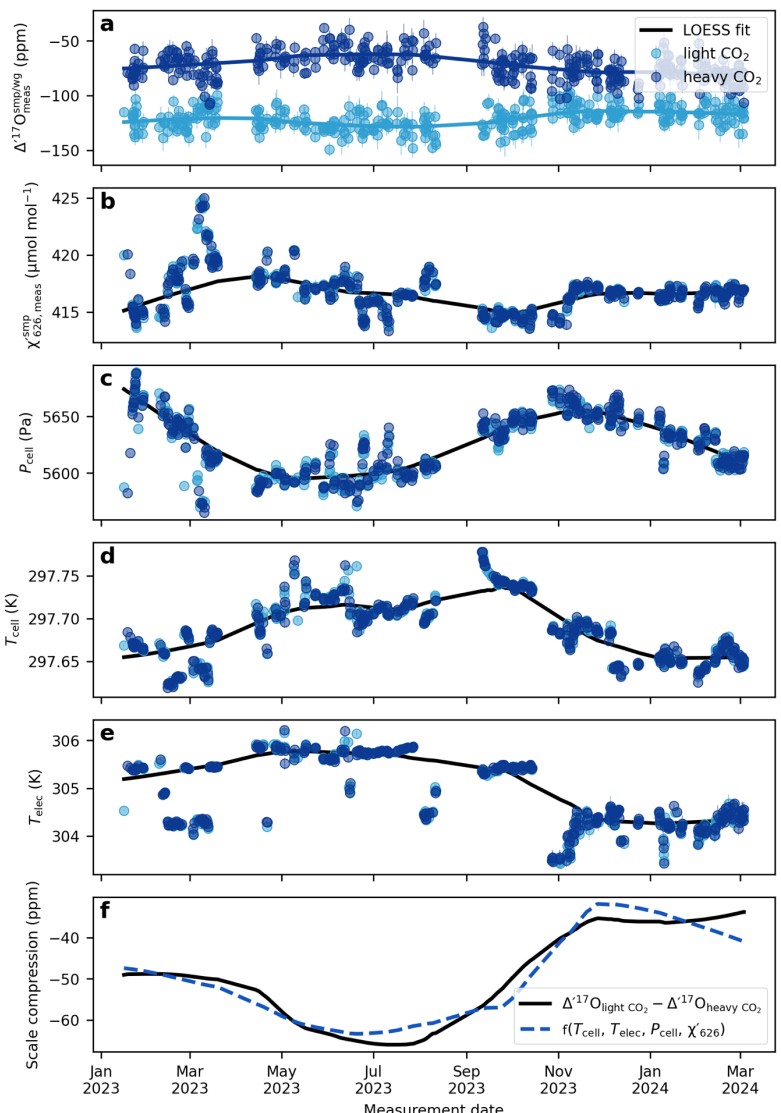

**Figure 7. The drift of the compression of the Göttingen setup.**

a) $\Delta'^{17}O$ values for the two internal reference gases, light $CO_2$ and heavy $CO_2$. b) average $\chi'_{626}$ of the sample cycles within a replicate. For the Göttingen setup the $\chi'_{626}$ of the reference cycles are within 1 µmol mol$^{-1}$ of the samples. c) cell pressure, d) cell temperature, e) electronics temperature, f) scale compression; defined as the difference between the $\Delta'^{17}O$ values of the light $CO_2$ and heavy $CO_2$. The solid lines depict LOESS fits to the data (smoothing: 0.4). The dashed blue line shows the multivariate linear regression model from the LOESS fits of the cell temperature, electronics temperature, cell pressure, and $\chi'_{626}$.

620

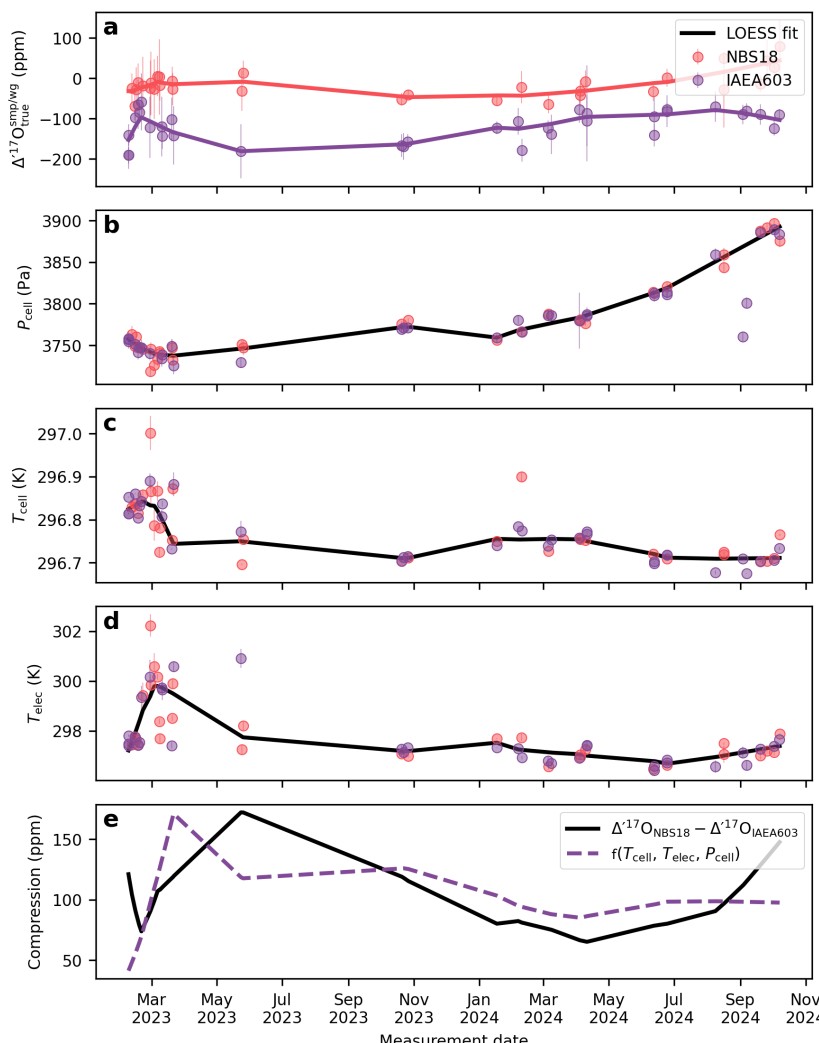

**Figure 8. The drift of the compression of the Cape Town setup.**

a) $\Delta'^{17}O$ values for two commonly used reference materials, NBS-18 and IAEA-603 b) cell pressure, c) cell temperature, d) electronics temperature, e) scale compression; defined as the difference between the $\Delta'^{17}O$ values of NBS-18 and IAEA-603. The dashed red line shows the multivariate linear regression model from the LOESS fits of the cell temperature, electronics temperature, and cell pressure.

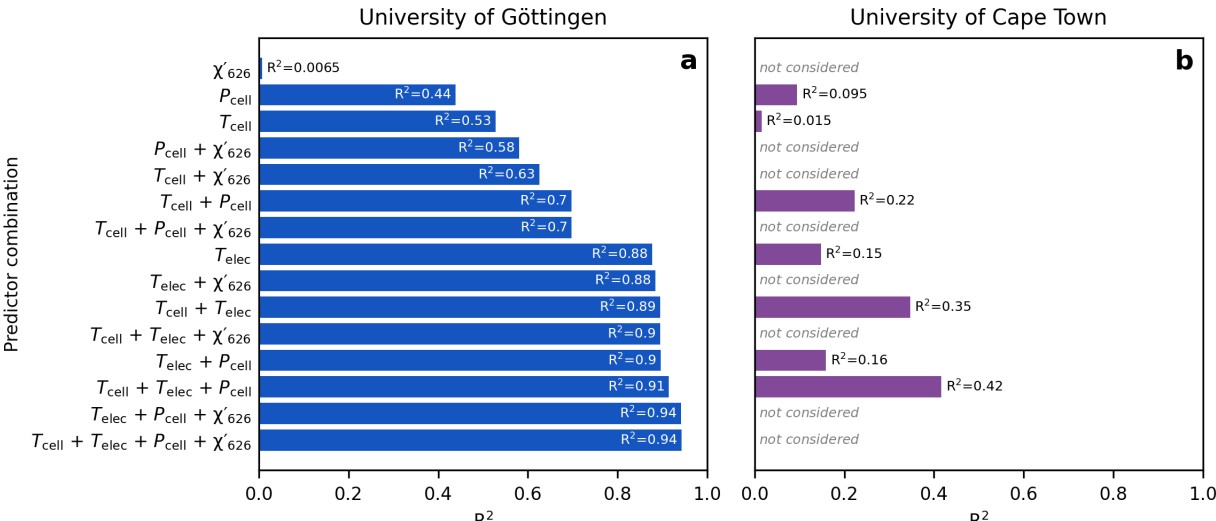

**Figure 9. Correlation coefficients of multiple linear regression models between scale compression in Δ'$^{17}$O and various combinations of cell and electronics temperature, cell pressure, and $\chi'_{626}$**

a) University of Göttingen, b) University of Cape Town. In this latter case, the models use only cell and electronics temperatures and cell pressure, as the Δ'$^{17}$O values were already corrected for scale-offset and normalized to a $\chi'_{626}$ value of 421 µmol mol$^{-1}$. The multiple linear regression models yielding the highest correlation coefficients are plotted on Figs 7f and 8e. We speculate that the comparatively weaker correlation in the Cape Town dataset is due to the fewer data points used to compute the LOESS fits. All $p$-values $< 0.05$