# Peer review of "Sensitivity of tunable infrared laser spectroscopic measurements of $\Delta^{17}$ O in CO2 to analytical conditions"

_EGUsphere, 2025_

## Author Response (AR1)

**Reviewer #1 - David Nelson**

We thank the reviewer for his constructive comments.

**The meaning of short term and long term in the abstract (and throughout the paper) are not well defined. The authors should define the time scale that they mean by long term drift.**

In the revised version, we explicitly state that by "long-term" we refer to a time span of several weeks. We also consistently use the phrase "long-term drift" throughout the manuscript. To avoid confusion, we omitted the phrase "short-term" and instead specify precisely what we mean, i.e., variations from one measurement cycle to the next within a replicate.

**Also it seems to me that pressure variation is also a short term drift. Since the pressure in the optical cell changes each time the cell is filled, it generally varies more rapidly than sample concentration. Hence, it would seem that pressure variation should also be categorized as a short term effect just as the authors do with variation in sample concentration.**

You are correct that pressure variations may contribute to cycle-to-cycle variations in the measured mixing ratios. In practice, however, the cell pressure is generally easy to maintain at a constant level, such that it does not measurably contribute to the cycle-to-cycle repeatability of the $\Delta'^{17}O$ values (Fig. 6c-d). In contrast, changes in analyte pressure over the course of several weeks, as shown in Figs 7, 8, and 9, do contribute to the long-term drift.

**Both concentration mismatch and pressure mismatch are drivers of instrumental measurement error. However, both are precisely quantified scalars whose effects can be quantified and corrected. The effects of temperature drift are much more complex beginning with the observation that there is not just one temperature. Many relevant temperatures are drifting simultaneously and continuously: cell temperature, laser temperature, the temperatures of various key electronic components, etc...**

Thank you very much for this comment! In response to your suggestion that there is not just one relevant temperature, we have also considered variations in the electronics temperature in the revised manuscript. Interestingly, we found that the variability in electronics temperature has a more significant effect on the drift than the cell temperature.

We added a corresponding discussion to the text and included electronics temperature in Figures 5–9.

**At line 113, I would suggest rephrasing to something like: "mixtures … are used to create optimal spectral line broadening due to collisional broadening at pressures between 30 and 40 Torr."**

Thank you for the comment! The revised sentence now reads "In TILDAS, mixtures of $CO_2$ and a collision gas (e.g., $CO_2$-free air or pure N2) are used to create optimal spectral line broadening due to collisional broadening at pressures between around 30 Torr and 40 Torr."

**In Section 3.2 the authors seem to give the impression that concentration dependence arises from the inability to measure all isotopologues of $CO_2$. I don't think this is correct. The root causes of concentration dependence are subtle and still being studied. But major factors seem to include systematic errors in the non-linear spectral retrievals and non-linearity in the infrared detector response function. Whatever the cause of concentration dependence, the empirical first order correction adopted by the authors (x=a*x' + b) is appropriate.**

You are correct, we rephrased the corresponding text and relate the root cause of concentration dependence to "systematic errors in the measurement of mole fractions and likely include nonlinearities related to spectral retrievals and the infrared detector response" (se beginning of Chapter 4.1).

**At line 255 perhaps it should be explicitly stated that the reference gas matrix must be selected to match the sample gas matrix.**

Very good point, thank you. We not only added this caveat but also expanded the corresponding section with two more paragraphs on additional discussion of gas-matrix–related effects (end of Chapter 4.3).

**At line 269 the authors state "Long-term drifts in analytical conditions — such as a gradual temperature change of 0.5 K over the course of a year". In some laboratories, temperature drifts of 0.5 K can occur within a few hours. There is a need to differentiate time drift versus temperature drift. Do the authors have data that isolate the effect of temperature on scale compression? That would be interesting to see. It**

**seems that a key question is: how frequently does a two point calibration need to be performed?  Or, perhaps, how much do such calibrations depend on ambient temperature?  The answer may depend on the range of 18O isotopic composition in the samples and standards being measured.**

A simple quantification is given in Chapter 4.2, where we note that in a not thermally stabilized setup a 2 K variation in the electronics temperature resulted in a ca 1500 ppm drift in $\Delta'^{17}O$. With the introduction of bracketing this is mitigated but the remainder temperature dependence depends on the speed of the speed of the temperature drift relative to the changeover time.

**Reviewer #2 – Mathieu Daëron**

**This kind of study, although perhaps non-glamorous, is extremely useful in the early days of a novel analytical technique. Virtually all groups already using optical methods to target cap-deltas in CO2 (or about to do so) will benefit from the observations reported here. Below, I list a number of comments/suggestions intending to improve and clarify the text.**

We thank the reviewer for his constructive comments.

**\* terms like "repeatability" "reproducibility" and "long-term stability" should be defined explicitly, early in the text. When using notions such as "standard deviation", it must be clear what "one observation" means (e.g., a single injection of an aliquot of CO2, the average of several injections, a one-second-long average optical measurement, etc). This is usually obvious to the authors but not to the readers, who may follow different conventions.**

In the revised version, we explicitly state what we mean by repeatability (i.e., whether it refers to 1 SE or 1 SD) and clearly indicate which subset of data each value pertains to.

**\* l. 43: Petersen et al. (2019) does not seem to be directly relevant here.**

Agreed! We removed the Petersen et al. (2019) reference.

**\* l. 48: Technically correct but somewhat misleading: the "precise" measurements reported by Adnew et al. (2019) come at the cost of prohibitively long integration times (20 h integration for 14 ppm internal SE).**

We added the caveat: "However, the above methods are generally too labor-intensive to be practical for routine monitoring of atmospheric $CO_2$."

**\* l. 60 "most apply to the technique in general": you may want to specify if "the technique" refers here to TILDAS of infrared spectroscopy in general. I am of course biased, but VCOF-CRDS as implemented by Chaillot et al. (2025) is not sensitive to analyte pressure, for example.**

We changed the word "technique" to "tunable laser absorption spectroscopy" to be clear.

**\* l. 153-154 "This indicates that the correction is independent of the isotopic composition of the sample analyte": what is the difference in $\delta^{13}C$, $\delta^{18}O$ values between IAEA-603-derived $CO_2$ and the working reference gas?**

In Göttingen the difference would be $\Delta\delta^{18}O_{CO2}$ = 10‰, whereas in Cape Town 25‰, however in Göttingen the light and heavy $CO_2$ differ by up to 48% from the $\delta^{18}O$ of the working gas. We do not discuss $\delta^{13}C$ in the paper. In the revised version we write: "The observation that gases with different isotope compositions ($\Delta\delta^{18}O$ relative to the working gas ranging from 28‰ to +48‰) yield identical correction slopes m within the same analytical sessions suggests that m is largely independent of the isotopic composition of the sample analyte."

**\* l. 156-157 "This suggests that the correction slope varies slightly between sessions": in practice, this means that one must determine lab-and-session-specific correction parameter(s) based on repeated analyses of some known CO2. These estimates for correction parameter(s) will have uncertainties, and it would seem useful to investigate/quantify the final contribution to analytical uncertainties from this source of error.**

Thank you for this comment. In the revised version, we now discuss the additional uncertainty introduced by the correction model (Eq. 9) to the data (see the end of Chapter 4.1 and Appendix A).

**\* l. 169 "eliminating the need for such a correction": to the authors' discretion, it might be relevant to note that this was tested and verified experimentally.**

Thank you for the remark!

**\* l. 176-182: Regarding the data shown in figure 5, it is not entirely clear if these measurements were corrected by repeated bracketing by a working standard, as reported in section 3.1. The next paragraph is quite confusing IMHO because of this: if the figure 5 data is reference-bracket-corrected, the δ18O and Δ17O variability seems enormous; if not, then what do these data look like after reference-bracket-correction? And if the corresponding reference gas measurements were not performed, how could the authors compute their "mismatch parameter"?**

In the revised version, we use consistent notation throughout the text and figures to clarify what each dataset refers to. Specifically, we use the subscript "smp/wg" for bracketed measurements and "meas" for raw values. In the case of Fig. 5, these are non-bracketed values. There was no changeover measurement performed during this test.

**\* figure 6: How is "internal $\Delta'^{17}O$ error" defined?**

In line 511, we write: "and the internal error of individual replicate measurements (68% confidence interval of the approximately 10 sample cycles bracketed by working gas analyses)". We also repeat this information in the caption of Fig. 6.

**\* l. 187-188 "The internal error of a single replicate analysis, i.e., the repeatability of approximately 10 sample cycles within a bracketing measurement": This is good example of ambiguity introduced by using undefined terms. Is the "internal error" not the standard error but the standard deviation of ~10 non-independent but separate bracket-corrected measurements? The answers to this question may be obvious for the authors, but not for most readers.**

See our comment above.

**\* l. 211-213 "This suggests that a temperature stability of ±1 mK, a pressure stability of ±10 Pa, and a χ'626 stability of ±1 µmol/mol are sufficient during a replicate measurements to prevent any systematic impact on internal repeatability.": I suggest quantifying this statement by adding "[any systematic impact on internal repeatability] beyond +/- X ppm on $\Delta'^{17}O$."**

Done!

**\* l. 220-221 "In this context, drift in compression directly affects the accuracy of the final $\Delta'^{17}O$ values.": This is surprising. One would expect that drift in compression would be effectively corrected by a two-anchor standardization approach. Is this not the case here?**

Thank you for this question as it prompted us to revisit this issue. On one hand, we rephrased the cited sentence to avoid ambiguity: "In this context, drift in the scale compression within a measurement period directly affects the accuracy of the final $\Delta'^{17}O$

values." On the other hand, we now note that "Figure 7a illustrates that the magnitude and direction of the drift in the Δ'17O values of the two standards used in the Göttingen laboratory are not identical. The δ18O values of these two standards differ by ca. 80‰. As discussed above and shown in Fig. 5, the mixing ratios of the three $CO_2$ isotopologues respond in an uncorrelated fashion to variations in analytical conditions related to systematic errors in the measurement of mole fractions. It follows that the magnitude of the drift in the measured δ and Δ'$^{17}$O values in response to changing analytical conditions, particularly temperature, depends on the isotopic composition of the analyte."

**\* l. 243-249: Here the authors make an important point, too often unacknowledged. Kudos for stating it succinctly and clearly.**

Thank you!

**\* l. 250-257: I would suggest adding recommendations about what to do in such cases. Is it feasible to inspect the fit residuals to look for a signature of such matrix effects? Would other approaches work better? Would labs using the exact same methods but different collision gases still get consistent results after two-point standardization?**

We expanded the paragraph on gas purity effects to provide additional details on the potential pitfalls of not matrix-matching the sample and working gas analytes. We now also state: "Matching the matrices of the working gas to that of the sample analyte as closely as possible helps prevent detrimental effects arising from variable scale-offsets." We note, however, that this may not be practical for air monitoring studies, where the analyte composition can vary (e.g., due to changing argon concentrations in air), and should therefore be further evaluated. One possible solution would be to extract $CO_2$ and dilute it again with the same collision gas used for the working gas.

**\* Regarding the monitoring of long-term drifts in analytical conditions, does this imply that things like temperature sensors should be periodically recalibrated to avoid drifts in true T (despite logged T being constant)?**

Calibrating sensors should be standard practice. In TDLWintel, the control software of Aerodyne Research Inc's TILDAS instruments, this is easily done under Edit->PTL...->Set P T offset. We added the following to our recommendations: "Continuous monitoring and reporting of the analytical conditions, along with the periodical recalibration of

temperature and pressure sensors, are therefore essential to ensure data integrity over extended timescales."